# Prognostic Factors for Post-COVID-19 Syndrome: A Systematic Review and Meta-Analysis

**DOI:** 10.3390/jcm11061541

**Published:** 2022-03-11

**Authors:** Giuseppe Maglietta, Francesca Diodati, Matteo Puntoni, Silvia Lazzarelli, Barbara Marcomini, Laura Patrizi, Caterina Caminiti

**Affiliations:** Clinical and Epidemiological Research Unit, University Hospital of Parma, University of Parma, 43126 Parma, Italy; gius.maglietta@gmail.com (G.M.); fdiodati@ao.pr.it (F.D.); mpuntoni@ao.pr.it (M.P.); slazzarelli@ao.pr.it (S.L.); bmarcomini@ao.pr.it (B.M.); lpatrizi@ao.pr.it (L.P.)

**Keywords:** long COVID, post-COVID-19 syndrome, prognostic factors, risk factors, meta-analysis

## Abstract

Evidence shows that a substantial proportion of patients with COVID-19 experiences long-term consequences of the disease, but the predisposing factors are poorly understood. We conducted a systematic review and meta-analysis to identify factors present during COVID-19 hospitalization associated with an increased risk of exhibiting new or persisting symptoms (Post-COVID-19 Syndrome, PCS). MedLine and WebOfScience were last searched on 30 September 2021. We included English language clinical trials and observational studies investigating prognostic factors for PCS in adults previously hospitalized for COVID-19, reporting at least one individual prospective follow-up of minimum 12 weeks. Two authors independently assessed risk of bias, which was judged generally moderate. Risk factors were included in the analysis if their association with PCS was investigated by at least two studies. To summarize the prognostic effect of each factor (or group of factors), odds ratios were estimated using raw data. Overall, 20 articles met the inclusion criteria, involving 13,340 patients. Associations were statistically significant for two factors: female sex with any symptoms (OR 1.52; 95% CI 1.27–1.82), with mental health symptoms (OR 1.67, 95% CI 1.21–2.29) and with fatigue (OR 1.54, 95% CI 1.32–1.79); acute disease severity with respiratory symptoms (OR 1.66, 95% CI 1.03–2.68). The I² statistics tests were calculated to quantify the degree of study heterogeneity. This is the first meta-analysis measuring the association between factors present during COVID-19 hospitalization and long-term sequelae. The role of female sex and acute disease severity as independent prognostic factors must be confirmed in robust longitudinal studies with longer follow-up. Identifying populations at greatest risk for PCS can enable the development of targeted prevention and management strategies. Systematic review registration: PROSPERO CRD42021253467.

## 1. Introduction

As the global COVID-19 pandemic progresses, it is becoming evident that a proportion of patients experiences multi-organ symptoms and complications lasting weeks or months after the acute phase of the disease. A systematic review and meta-analysis of 15 studies and a total of 47,910 patients [1] measured the prevalence of more than 50 persistent and new patient-reported symptoms, including chronic cough, shortness of breath, chest tightness, cognitive dysfunction and extreme fatigue. In another systematic review of 57 studies comprising more than 250,000 survivors of COVID-19 [2], more than half experienced post-acute sequelae six months after recovery. The implications and consequences of such ongoing clinical manifestations, often referred to as long COVID, are a growing concern for health, and represent a major challenge for health care systems. This makes it of paramount importance to understand the factors predisposing the development of long-term consequences, enabling us to identify vulnerable individuals and help health authorities to prepare for early screening and diagnosis, as well as to establish proper facilities to care for their needs. For clinicians, identifying populations at greatest risk for long COVID will facilitate the timely provision of appropriate interventions and support, with whole-patient perspectives aimed at reducing morbidity and improving outcomes [1,3]. Despite the crucial importance of understanding the risk factors for COVID-19 sequelae, the literature so far has mainly focused on the prevalence of persistent signs and symptoms. Some literature reviews mention prognostic aspects [4,5,6,7,8,9], but mainly as narrative descriptions without data synthesis, and including heterogeneous studies (considering different populations, e.g., with and without prior COVID-19 hospitalization, and with different follow-up durations). We therefore carried out this systematic review and meta-analysis to identify, in patients who had been hospitalized for COVID-19, which factors already present or emerging during hospitalization, were associated with an increased risk of exhibiting new or persisting symptoms.

In this review, we will refer to Post-COVID-19 Syndrome (PCS) to mean “signs and symptoms that develop during or after an infection consistent with COVID-19, continue for more than 12 weeks and are not explained by an alternative diagnosis”, as defined by the National Institute for Care and Excellence (NICE) in its “COVID-19 rapid guideline” [10].

## 2. Materials and Methods

Before conducting this work, in April 2021, the PROSPERO database [11] was searched to identify any existing review on the subject to avoid replication, but none was found. This review was designed and conducted following the Preferred Reporting Items for Systematic reviews and Meta-Analyses (PRISMA) guidelines [12], and the indications of Riley et al. specific to systematic reviews and the meta-analysis of prognostic factor studies [13]. The protocol was registered with PROSPERO (CRD42021253467) on 14 May 2021.

### 2.1. Search Strategy

Studies were identified by searching the MEDLINE database using the PubMed platform, and Web Of Science Clarivate, with no date restriction. The search was first performed on 20 July 2021 on PubMed, and then rerun on September 30th on both databases. A backwards snowball search of the references of systematic reviews was also conducted. The full search terms and notes on strategy development are given in the online Appendix (Appendix A).

### 2.2. Eligibility Criteria

Study eligibility: clinical trials and observational studies (cohort studies, case control, cross-sectional studies, historic cohort studies) were considered. We excluded reviews, editorials, commentaries, methodological articles, letters to editors, and case reports, along with duplicates/replicates of studies. To be included, studies had to investigate prognostic factors for PCS in consecutive samples of adult patients, reporting at least one individual prospective follow-up at 12 weeks or later. The minimum follow-up length allowed us to exclude the short-term effects of the disease.

Population eligibility: the review concerned adult patients discharged after having been hospitalized for COVID-19. Research on individuals admitted to an Emergency Department but not to a hospital ward was not considered. This criterion was applied in order to identify risk factors present during hospitalization which may be promptly recognized and used for care decisions. Studies of minors were excluded because at the time this review was conducted they made up a small proportion of COVID-19-affected individuals. Only publications in the English language were reviewed.

### 2.3. Selection Process

Two reviewers independently performed initial title and abstract screening for relevance to the review, using the Rayyan platform [14]. Publications not dealing with PCS or clearly stating that follow-up was less than 12 weeks were excluded at this stage. Next, two reviewers independently examined the full texts of the screened publications and performed the study selection. At this stage, reviewers ascertained the presence of: (1) individual follow-up of at least 12 weeks to detect signs and symptoms attributable to COVID-19; (2) the analysis of prognostic factors present or emerged during hospitalization, preferably included in a multivariate model. Any disagreements were resolved by a third, independent, reviewer.

### 2.4. Data Extraction

Data items that were extracted included: title and first author, study design, objectives, mono- or multi-center, patient characteristics, number of included patients, follow-up mode, follow-up length, sequelae and symptoms of PCS, risk factors, estimates of the prognostic effect, and measures of variability. Information was extracted by two reviewers using a Microsoft Excel form. Discrepancies were resolved through discussion and, when necessary, by involving a third reviewer. Some study investigators were contacted when data confirmation was needed. Once data were extracted, the two reviewers assessed the applicability of each study in the review, with any discrepancies resolved by the third reviewer. Applicability refers to the extent to which a selected study matched the review question in terms of the population, timing, prognostic factors, and outcomes of interest [13]. Because of the heterogeneity in prognostic factor studies, more deviations from the defined question are possible (far greater than what is typically encountered during the selection of randomized intervention studies) [13].

### 2.5. Risk of Bias Assessment

The risk of bias (methodological quality) in the included studies was assessed using the QUIPS tool (Quality in Prognosis Studies) [15] which is recommended by the Cochrane Prognosis Methods Group, has been used in several reviews [16,17,18] and has acceptable inter-rater reliability. The QUIPS checklist appraises six domains: study participation, study attrition, prognostic factor measurement, outcome measurement, adjustment for other prognostic factors, and statistical analysis and reporting. Risk of bias is scored as low, moderate or high. Two reviewers independently applied the tool to each included study, recording the supporting information and justifications for judgment of risk of bias for each domain. Doubts were resolved through discussion. We assessed risk of bias for the study overall, rather than separately by outcome measure. Studies were judged as having an overall low risk of bias if all the six domains had been rated as low.

### 2.6. Data Synthesis

To summarize the prognostic effect of each factor (or group of factors) of interest, the odds ratios were estimated using raw data. A meta-analysis of adjusted results was not possible because included studies were exploratory, i.e., aimed at identifying associations between a number of potential prognostic factors and a set of outcomes [19]. Furthermore, most studies used different methods of measurement and categorization for prognostic factors and outcomes and various sets of adjustment factors, and did not report a multivariable analysis. We decided to perform a meta-analysis, in spite of these limitations, to provide hypothesis-generating evidence indicative of a potential association between a prognostic factor and an outcome [19].

We performed random-effects meta-analyses using the Paule and Mandel method for the estimation of between-study variance [20,21]. The confidence intervals of the overall effects of prognostic factors on PCS were adjusted applying the Hartung-Knapp-Sidik-Jonkman (HKSJ) approach [13,22,23,24] to account for the uncertainty in the variance estimates. The I² statistics tests were calculated to quantify the degree of study heterogeneity [25]. The I² value to establish either low or high heterogeneity was 30%. The level of significance was set at *p* < 0.050.

We performed subgroup analyses to explore the causes of heterogeneity for outcomes composed by multiple symptoms. Furthermore, although we planned to assess publication bias for each meta-analysis including ≥10 studies by funnel plot representation and a Peter’s test at a 10% level, none of them met this criterion.

Data were processed using R statistical software (R: a language and environment for statistical computing.The R Foundation for Statistical Computing, Vienna, Austria), v. 4.0.3, with the meta and metasens packages [26].

### 2.7. Patient and Public Involvement

Patients and the public were not involved in this research.

## 3. Results

### 3.1. Study Selection

A total of 1614 articles were retrieved from the two databases and uploaded into the Rayyan platform. Duplicates and ineligible publication types according to above-mentioned criteria (995 records) were excluded automatically. Title and abstract screening of the remaining 619 records identified 117 potentially eligible articles, which underwent full text review. A total of 97 studies were excluded following the applicability assessment (Figure 1) and twenty studies were eventually included in the review and meta-analysis [27,28,29,30,31,32,33,34,35,36,37,38,39,40,41,42,43,44,45,46].

No additional paper was manually identified from the reference lists of the systematic reviews. A total of 14 studies were identified by the first PubMed search on July 20th [27,28,31,32,33,34,35,36,38,39,41,42,43,44], and six were identified by the search rerun on September 30th [29,30,37,40,45,46].

A flow diagram depicting the selection process is provided in Figure 1.

### 3.2. Study Characteristics

The characteristics of the 20 included studies are shown in Table 1. In total, 13,340 patients were considered, of whom 6213/13,051 (47.6% in 19 studies) were women. Most studies were set in Europe (11/20, 55%) [27,28,29,30,31,32,36,37,38,40,42], followed by China (8/20, 40%) [33,34,35,39,43,44,45,46], while one study was conducted in South America [41]. Nine of the twenty studies were multicenter [28,29,30,32,37,38,39,45,46]. The majority were ambidirectional, concerning patient cohorts or case series, where data relating to the acute phase (risk factors) were collected retrospectively, and the presence of PCS (outcome) was recorded prospectively. Eight of twenty studies [28,29,31,32,34,41,43,44] had a longitudinal design with one or more follow-up time points. Follow-up was conducted through outpatient visits in ten studies [28,29,31,32,33,34,35,42,43,46], and through telephone interviews in nine [27,30,36,37,39,40,41,44,45]; only one study used both modalities [38].

### 3.3. Quality of Included Studies

Using the QUIPS tool, 11 studies [27,28,29,31,34,35,41,42,44,45,46] scored high in at least one domain, and nine [30,32,33,36,37,38,39,40,43] scored moderate in at least one domain (Appendix A). The most frequent high risk of bias, detected in ten of the twenty articles, concerned study attrition: i.e., the representativeness of participants with follow-up-data with respect to those enrolled in the study (selection bias).

### 3.4. Factors Associated with PCS Risk

Risk factors were included in the analysis if their association with PCS was investigated by at least two studies. Given the large heterogeneity between studies concerning the investigated associations and the small sample sizes, we decided to focus our analysis on the most commonly reported predictors (i.e., sex and disease severity during the acute phase), and, consequently, only on the considered outcomes: any symptoms, respiratory symptoms, mental health symptoms, fatigue. All 20 studies contributed at least once to the estimate of the association between risk factor and outcome. Regarding severity, we considered any study which classified patients according to severity of COVID-19 during hospitalization from national or international classification systems, to symptom characteristics and length of stay, to ICU use (Appendix A), although the criteria used varied greatly.

The results of the meta-analysis are shown in Figure 2, Figure 3, Figure 4, Figure 5, Figure 6 and Figure 7.

#### 3.4.1. Any Symptoms 

For this outcome (Figure 2), the association with sex was investigated by eight out of 20 studies [29,31,32,33,37,38,39,45], including 9421/13,340 (70.6%) patients, 47.7% female (4494/9421). The occurrence of any symptoms was 56.3% in women (2531/4494) and 45.5% in men (2241/4927). All studies reported an OR > 1 for female sex (range 1.15–2.00), and the overall OR was statistically significant (OR 1.52; 95% CI 1.27–1.82).

#### 3.4.2. Respiratory Symptoms 

The association of at least one respiratory outcome with sex (Figure 3) was investigated by twelve of twenty studies [27,28,30,33,34,35,37,38,39,43,44,45] including 10,874/13,340 (81.5%) patients. Romero-Duarte et al. [38] contributed to six different respiratory outcomes, Zhang et al. [45] contributed to four, Shang et al. [39] contributed to three, Huang et al. [33] contributed to two, and the remaining [27,28,30,34,35,37,43,44] to one. Two studies [37,38] investigated any respiratory symptoms, while the remaining studies analyzed specific symptoms, such as dyspnea, cough and shortness of breath. Five studies found a statistically significant association, of which one [44] was found with male sex, and four [27,30,37,43] were found with female sex. Considering individual symptoms, Diffusing Lung Capacity for Carbon Monoxide (DLCO) less than 80% and sore throat exhibited a large OR close to statistical significance (OR 2.28, 95% CI 0.99–5.27 and OR 1.40, 95% CI 0.94–2.07, respectively). The overall analysis, though characterized by substantial statistical heterogeneity (I^2^ 65%, *p* < 0.01), confirmed a tendency towards significance for the association between respiratory symptoms and female sex (OR 1.20, 95% CI 1.00–1.45). The subgroup analysis confirmed high heterogeneity (I^2^ 30–87%) for all individual symptoms, except for sore throat.

Figure 4 shows the associations with acute disease severity explored by nine of the twenty studies (45%) [27,28,34,35,36,39,42,44,46], on 2530/13340 (19%) patients. Two studies [36,39] examined cough, five [27,28,35,42,46] examined DLCO, and the remaining [34,44] examined other respiratory symptoms. In the subgroup analysis, cough and DLCO were statistically significantly associated (OR 1.78, 95% CI 1.05–3.03; OR 2.05, 95% CI 1.06–3.96, respectively) with acute disease severity (severe or critical). The pooled estimate was also statistically significant (OR 1.66, 95% CI 1.08–2.57), although it exhibited high heterogeneity (I^2^ 71%).

#### 3.4.3. Mental Health Symptoms 

Seven studies (35%) with 6383/13,340 (47.9%) patients [27,33,38,39,40,41,45] included in the review examined the influence of sex on mental health, as shown in Figure 5. The overall analysis showed the unfavorable effect of female sex (OR 1.67, 95% CI 1.21–2.29). The subgroup analysis highlighted the weight of the individual symptom anxiety, which investigated by three studies [38,41,45] including a total of 1693/3465 women (48.9%), and which was statistically influenced by being female (OR 1.95, 95% CI 1.52–2.49), with low heterogeneity between studies (I^2^ 8%).

#### 3.4.4. Fatigue 

We found seven large studies (35%) [30,33,37,38,39,44,45], including 10,088/13,340 (75.6%) eligible patients which examined the influence of female sex (4895/8724 women included in the analysis, 56.1%). This association was statistically significant, with OR 1.54, 95% CI 1.32–1.79 (Figure 6).

As shown in Figure 7, acute disease severity had no statistically significant effect on fatigue, although with OR of 1.23 (95% CI 0.73–2.07). This association was investigated by four of the twenty (25%) studies [39,44,45,46] including 3861/13,340 patients (28.9%).

## 4. Discussion

### 4.1. Main Findings

Our meta-analysis of 20 studies and 13,340 patients found that two factors, being female and having a severe clinical picture during hospitalization, increased the risk of PCS. These results should be interpreted with caution, as they were obtained by pooling crude estimates, and thus need further verification. While the association with acute disease severity can be explained more easily, the influence of female sex requires further investigation, especially in consideration of the opposite influence found during the acute phase. In fact, a recent meta-analysis of 55 studies [47] found that males were more likely to be infected with COVID-19 and go into serious condition (OR = 2.41, *p* < 0.00001) than females, confirming what emerged from previous reviews [48,49]. Proposed explanations for these differences include the protective effect of the X chromosome and sex hormones, which play an important role in innate and adaptive immunity, and the stronger IgG antibody production found in early-stage women [47,50]. Other hypothesized reasons are related to the higher mortality of males during the acute phase. As cautioned by Di Toro and colleagues [51], the estimates concerning long COVID do not reflect the epidemiology of COVID-19, but only that relating to COVID-19 survivors, thus excluding the high number of deceased patients, mainly older males.

### 4.2. Strengths and Weaknesses of the Study

To our knowledge, this is the first systematic review and meta-analysis measuring the association between factors present during COVID-19 hospitalization and long-term sequelae. Other reviews of the literature mentioning risk factors for PCS have as their main question the measure of prevalence, and they summarize the potential prognostic factors only in a narrative way, without making a pooled estimate of the effects [4,5,6,7,8,9]. This meta-analysis was carried out following existing recommendations [13,15]. In particular, we performed the HKSJ procedure, which yields a wider and more rigorous confidence interval, and used a validated tool to examine each study’s risk of bias.

The studies were generally of moderate quality. The most frequent problem was related to possible selection errors due to large numbers of people lost to follow-up and/or the lack of comparison between participants and non-participants. Half of the examined studies were conducted in one center only, none indicated a priori sample size calculation, and identification of prognostic factors was almost always a secondary objective. Notably, only one study [44] used a control group. These observations seem to confirm the less-than ideal quality levels of much of COVID-19 research, which is often performed in challenging conditions of urgency [52].

Heterogeneity represents another important limitation of the evidence analyzed in this paper. The wide range of considered risk factors and the use of different definitions, classifications and measurement tools for detecting the same symptoms forced us to exclude some outcomes and risk factors of potential interest from the meta-analysis. In any case, for the included studies, heterogeneity was addressed by applying the random-effects model and the Hartung-Knapp correction [24].

Other limitations of this work should be acknowledged. Firstly, we realize that relevant papers may have been missed due to the difficulty of identifying prognostic studies. In fact, compared to randomized trials of interventions, prognostic studies do not tend to be indexed (“tagged”) because a taxonomy of prognosis research is not widely recognized, and there is much more variation in their design [13]. To minimize this problem, our search strategy was constructed using published approaches designed to retrieve specific types of records (see online appendix in Appendix A). Secondly, we restricted eligibility to publications in English, which may have overlooked relevant publications written in other languages.

### 4.3. Implications for Future Research 

Our systematic review and meta-analysis offers information on a set of variables which should be considered for prognostic factor selection in multivariable modeling in further study designs. It also contributes to identifying knowledge gaps. The many limitations of the current evidence found in this review emphasize the need for confirmatory prognosis studies which test independent prognostic factors with longitudinal design and periodic (at least six-monthly) prospective long-term follow-ups (at least for 3 years), of adequate size for statistical analysis, with combined control groups. It is also necessary to standardize how each sign and symptom of the PCS should be defined and measured in future research. These studies should lead to the development of a predictive model used for the timely identification on individuals at risk of long COVID, similar to the model explored by Sudre et al. [53].

### 4.4. Implications for Clinical Practice

Since this work is exploratory in nature, it cannot provide indications for practice, which should be based on robust, confirmatory research. It is becoming evident that healthcare systems around the world will soon be overwhelmed by the need to provide long-term care for COVID-19 survivors suffering from sequelae of the illness [6]. For a long time, patients have been raising awareness of the persistent physical and psychosocial health consequences of COVID-19, making heterogeneous and complex symptoms collectively visible that were not commonly recognized by the scientific community [54]. To respond effectively to patient needs and concerns, sound data are urgently needed to guide the development of programs and infrastructures for the prevention and management of long COVID. Knowing which types of populations are at greatest risk of long-term effects will enable efforts to be aimed at the people who could ultimately benefit from them, thus maximizing available resources.

## 5. Conclusions

This systematic review and meta-analyses indicates that female sex and acute disease severity may predispose patients to long-term symptoms of COVID-19. Studies on this topic exhibit multiple methodological issues, and large amount of variability exists between studies in terms of considered risk factors, definitions, follow-up duration and modality, etc. Rigorous research in this field should be encouraged, guided by specific scientific recommendations, to help healthcare systems face the needs of the growing number of individuals who may require long-term care after infection.

## Figures and Tables

**Figure 1 jcm-11-01541-f001:**
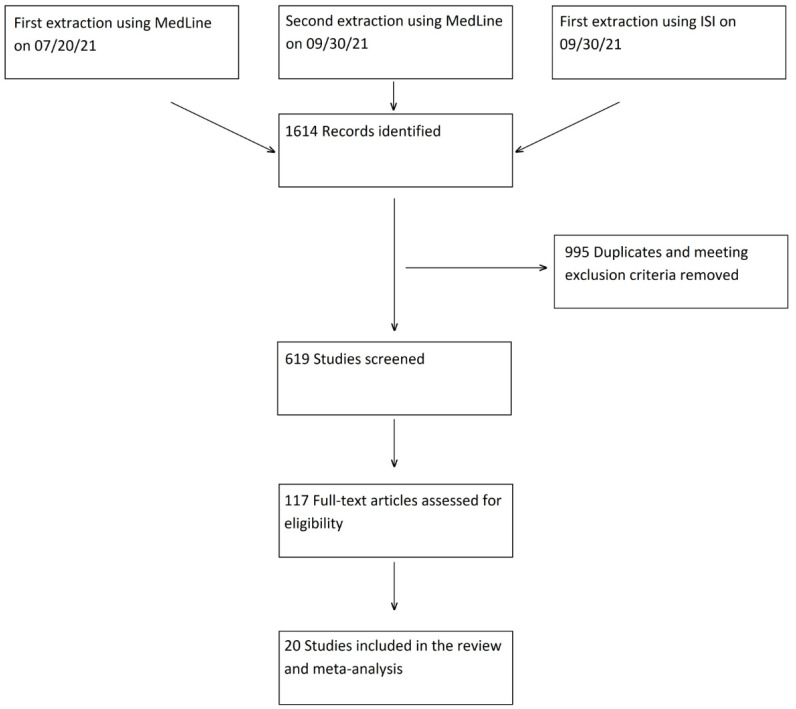
Flow diagram.

**Figure 2 jcm-11-01541-f002:**
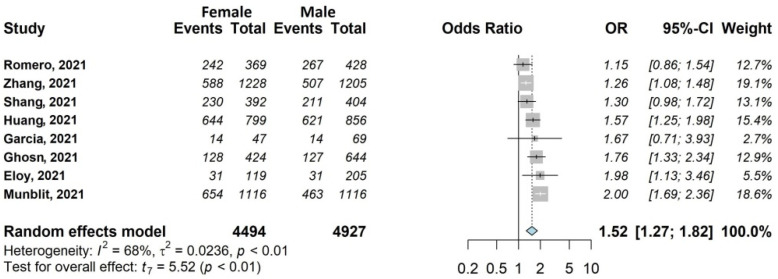
Forest plots of adjusted analyses for association between sex (female) and any Symptoms. HKSJ, Hartung-Knapp-Sidik-Jonkman.

**Figure 3 jcm-11-01541-f003:**
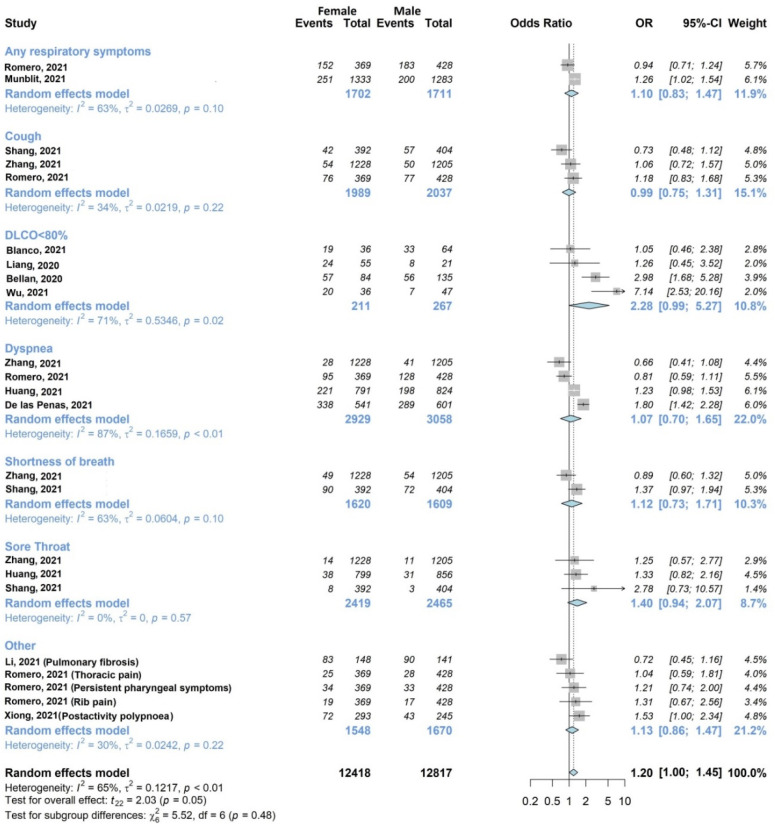
Forest plots of adjusted analyses for association between sex (female) and respiratory symptoms. HKSJ, Hartung-Knapp-Sidik-Jonkman.

**Figure 4 jcm-11-01541-f004:**
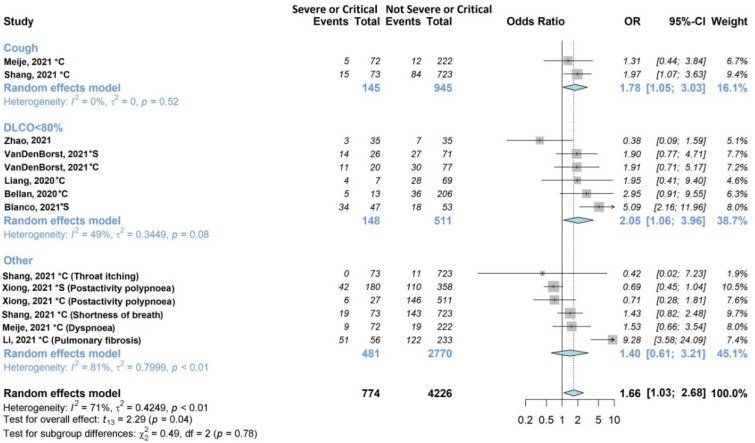
Forest plots of adjusted analyses for association between acute disease severity and respiratory symptoms. HKSJ, Hartung-Knapp-Sidik-Jonkman.

**Figure 5 jcm-11-01541-f005:**
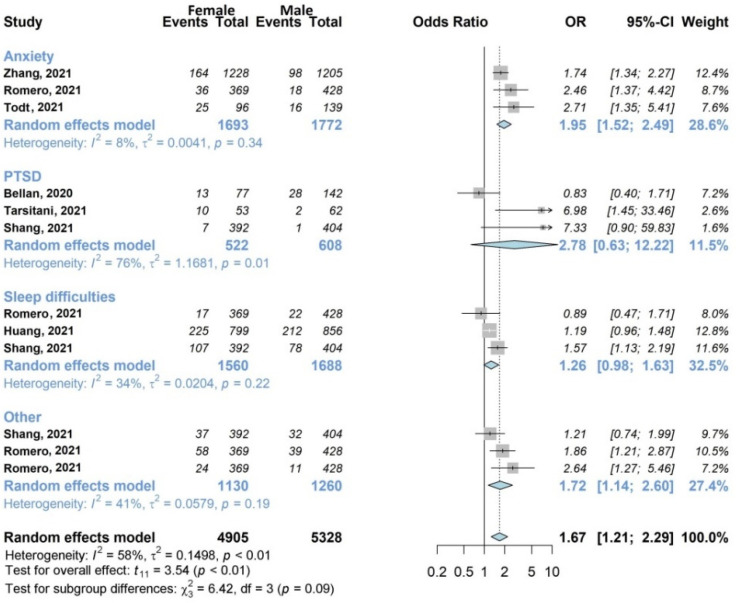
Forest plots of adjusted analyses for association between sex (female) and mental health. HKSJ, Hartung-Knapp-Sidik-Jonkman.

**Figure 6 jcm-11-01541-f006:**
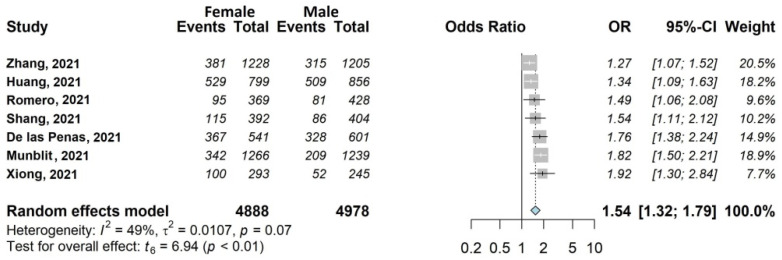
Forest plots of adjusted analyses for association between sex (female) and fatigue. HKSJ, Hartung-Knapp-Sidik-Jonkman.

**Figure 7 jcm-11-01541-f007:**
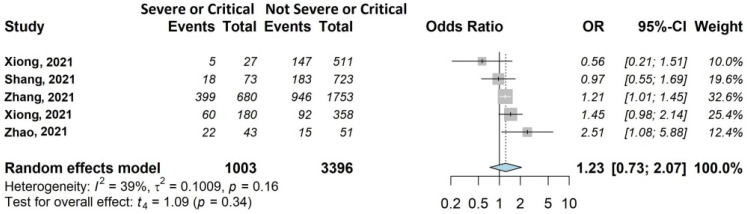
Forest plots of adjusted analyses for association between acute disease severity and fatigue. HKSJ, Hartung-Knapp-Sidik-Jonkman.

**Table 1 jcm-11-01541-t001:** Characteristics of included studies.

First Author	Study Design	Country	Covid Patients Included, No	Age (Years)	Sex (% Female)	Follow-Up Length	Follow-Up Mode
Bellan et al. [27]	Ambidirectional study, single center	Italy	238	Median (IQR): 61 (50–71)	96 (40.3%)	3–4 months after discharge	Telephone interview and outpatient visit
Blanco et al. [28]	Longitudinal, prospective study, multicenter	Spain	100	>50 years: 69 (69%)	36 (36%)	Median (IQR): 104 (89.25–126.75) after onset	Outpatient visit
Eloy et al. [29]	Longitudinal, prospective study, multicenter	France	324	Median (IQR): 61 (52–69)	119 (37%)	6 months after discharge	Outpatient visit
Fernández-de-Las-Peñas et al. [30]	Ambidirectional study, multicenter	Spain	1142	Mean (SD): 61 + 17	548 (48%)	7 months after discharge	Telephone interview
García-Abellán et al. [31]	Longitudinal, prospective study, single center	Spain	146	Median (IQR): 64 (54–76)	58 (40%)	6 months after discharge	Outpatient visit
Ghosn et al. [32]	Longitudinal, prospective study, multicenter	France	1137	Median (IQR): 61 (51–71)	424 (37%)	6 months after admission	Outpatient visit
Huang et al. [33]	Ambidirectional study, single center	China	1733	Median (IQR): 57 (47–65)	836 (48%)	Median (IQR): 186 (175–199) days after onset	Outpatient visit
Li et al. [34]	Longitudinal, prospective study, single center	China	289	Not available	Not available	From 90 to 150 days after onset	Outpatient visit
Liang et al. [35]	Ambidirectional study, single center	China	76	Mean (SD): 41.3 ± 13.8	55 (72%)	3 months after discharge	Outpatient visit
Meije et al. [36]	Ambidirectional study, single center	Spain	302	Mean (SD): 68.8 (12.7)	131 (43%)	7 months after discharge	Telephone interview
Munblit et al. [37]	Ambidirectional study, multicenter	Russia	2649	Median (IQR): 56 (46–66)	1353 (51.1%)	6–8 months after discharge	Telephone interview
Romero-Duarte et al. [38]	Ambidirectional study, multicenter	Spain	797	Mean (SD): 63 (14.4)	369 (46.3%)	6 months after discharge	Outpatient visit and Telephone interview
Shang et al. [39]	Ambidirectional study, multicenter	China	796	Median (IQR): 62 (51–69)	392 (49.2%)	6 months after discharge	Telephone interview
Tarsitani et al. [40]	Ambidirectional study, single center	Italy	115	Median (IQR): 57 (48–66)	67 (58%)	3 months after discharge	Phone interview
Todt et al. [41]	Longitudinal, prospective study, single center	Brazil	251	Mean (SD): 53.6 (+14.9)	101 (40.2%)	3 months after discharge	Telephone interview
van den Borst et al. [42]	Ambidirectional study, single center	The Netherlands	97	Not available	31 (32%)	Mean (SD): 13 (2.2) weeks after onset	Outpatient visit
Wu et al. [43]	Longitudinal, prospective study, single center	China	83	Median (IQR): 60 (52–66)	36 (43%)	3, 6, 9, and 12 months after discharge	Outpatient visit
Xiong et al. [44]	Longitudinal, prospective, controlled study, single center	China	538	Median (IQR): 52 (41–62)	293 (54.5%)	Median (IQR): 97 (95–102) days after discharge	Telephone interview
Zhang et al. [45]	Ambidirectional study, multicenter	China	2433	Median (IQR): 60 (49–68)	1228 (50.5%)	1 year after discharge	Telephone interview
Zhao et al. [46]	Ambidirectional study, multicenter	China	94	Mean 48.11	40 (42.55%)	1 year after discharge	Outpatient visit

## Data Availability

Data is contained within the article or Appendix A.

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
