# Peer review of "Prognostic Factors for Post-COVID-19 Syndrome: A Systematic Review and Meta-Analysis"

_jcm, 2022, doi:10.3390/jcm11061541_

Round 1
Reviewer 1 Report
It is an interesting manuscript that synthesizes evidence on a condition of general interest for global public health, considering the burden of disease that post-COVID 19 syndrome is generating. I kindly request the following corrections. After that, I suggest that the manuscript be accepted:
- There is an inconsistency between the search start date in the summary methods section and that found in the methods section.
- Please correct figures 2b, 2c, and 2d, if you have already placed the title of the event evaluated, it is not necessary to repeat it. Only under the evaluated event, place the surname and the year or the reference (be guided by Cochrane reviews).
The discussion and justification presented by the authors are adequate and of high quality. The methodology is clear and reproducible. It is a systematic review that complies with the items of the PRISMA guidelines. Please make the mentioned corrections
Reviewer 2 Report
Title: Prognostic factors for Post-Covid-19 Syndrome: a systematic review and meta-analysis “Stage 1” - Giuseppe Maglietta , Francesca Diodati , Matteo Puntoni , Silvia Lazzarelli , Barbara Marcomini , Laura Patrizi , Caterina Caminiti *
Journal of Clinical Medicine - jcm-1592629
In this meta-analysis, Maglietta et al. have tried to identify associated risk factors for COVID related complications. This is an interesting manuscript with relevance to the current pandemic. Here are my comments:
- In the abstract, the Authors have mentioned the risk of bias at two places. The authors should combine both sentences. 'Two authors independently assessed risk of bias.' and 'Risk of bias was generally moderate.' Refer following segment 'weeks. Two authors independently assessed risk of bias. 22 Risk factors were included in the analysis if their association with PCS was investigated 23 by at least two studies. To summarize the prognostic effect of each factor (or group of 24 factors), odds ratios were estimated using raw data. Overall, 20 articles met the inclusion 25 criteria, involving 13,340 patients. Risk of bias was generally moderate.'
- In the abstract, authors can restructure the sentence to emphasize on the variables, for example readers should know the 'factor' to which they are predisposed 'The following 26 associations were statistically significant: Female sex and Any symptoms (OR 1.52; 27 95%CI 1.27-1.82), Acute Disease Severity and Respiratory symptoms (OR 1.66, 95%CI 28 1.03-2.68), Female sex and Mental Health symptoms (OR 1.67, 95%CI 1.21-2.29), Female 29 sex and Fatigue (OR 1.54, 95%CI 1.32-1.79).'
- For the readers of the journal, authors can expand more on the risk of bias and cite references. Authors can also cite other meta-analyses where these methods have been utilized.
Refer-'Risk of bias is scored as low, moderate, or 140 high. Two reviewers independently applied the tool to each included study, 141 recording supporting information and justifications for judgment of risk of bias 142 for each domain. Doubts were resolved through discussion.'
- Was the data normalized or controlled for batch effects? If yes, they should be mentioned, and if no, an appropriate discussion should be added. Refer'To summarizes the prognostic effect of each factor (or group of factors) of 145 interest, the odds ratios were estimated using raw data. Meta-analysis of ad-146 justed results was not possible because included studies were exploratory, i.e. 147 aimed at identifying associations between a number of potential prognostic 148 factors and a set of outcomes'
- Authors must provide references where these methods were utilized. Refer these methods: 'Paule and Mandel 155 method for the estimation of between-study variance' ,'The confidence in-156 tervals of the overall effects of prognostic factors on PCS were adjusted apply-157 ing the Hartung-Knapp-Sidik-Jonkman (HKSJ) approach', and 'The I² statistics tests were calculated to 159 quantify the degree of study heterogeneity. The I² value to establish either low 160 or high heterogeneity was 30%. The level of significance was set at p < 0.050.'
- Authors should cite authors of these packages. They can provide references to the original research paper. Refer section 'Data were processed using R statistical software, v. 4.0.3 with meta and 167 metasens packages.'
- These are: Schwarzer, G., Carpenter, J.R. and Rücker, G., 2015. Meta-analysis with R (Vol. 4784). Cham: springer.
- Authors can add a conclusion section: They can start with the question, methodology used, crystallize their main findings, and succinctly provide future direction.
- Authors can enrich their references with the following important publications:
- Sudre, C.H., Murray, B., Varsavsky, T., Graham, M.S., Penfold, R.S., Bowyer, R.C., Pujol, J.C., Klaser, K., Antonelli, M., Canas, L.S. and Molteni, E., 2021. Attributes and predictors of long COVID. Nature medicine, 27(4), pp.626-631.
- Nalbandian, A., Sehgal, K., Gupta, A., Madhavan, M.V., McGroder, C., Stevens, J.S., Cook, J.R., Nordvig, A.S., Shalev, D., Sehrawat, T.S. and Ahluwalia, N., 2021. Post-acute COVID-19 syndrome. Nature medicine, 27(4), pp.601-615.
Round 2
Reviewer 2 Report
The authors have significantly improved the manuscript.